# Effect of Chemokine (C-C Motif) Ligand 7 (CCL7) and Its Receptor (CCR2) Expression on Colorectal Cancer Behaviors

**DOI:** 10.3390/ijms20030686

**Published:** 2019-02-05

**Authors:** Ewa Kurzejamska, Mariusz Sacharczuk, Natalia Landázuri, Oksana Kovtonyuk, Marzena Lazarczyk, Sharan Ananthaseshan, Zbigniew Gaciong, Piotr Religa

**Affiliations:** 1Department of Medicine, Centre for Molecular Medicine, Karolinska Institute, 171 76 Stockholm, Sweden; natalia.landazuri@ki.se (N.L.); sharan.anand23@gmail.com (S.A.); piotr.religa@ki.se (P.R.); 2Department of Internal Medicine, Hypertension and Vascular Diseases, Medical University of Warsaw, 02 097 Warsaw, Poland; m.sacharczuk@ighz.pl (M.S.); kseniya_kovtonyuk@yahoo.com (O.K.); marzena.lazarczyk@gmail.com (M.L.); zbigniew.gaciong@wum.edu.pl (Z.G.); 3Laboratory of Neurogenomics, Institute of Genetics and Animal Breeding, Polish Academy of Sciences, 05 552 Jastrzębiec, Poland; 4Institute of Health Sciences, Pope John Paul II State School of Higher Education, 21-500 Biała Podlaska, Poland

**Keywords:** chemokine, CCL7, colon cancer, tumor growth, metastasis, immunoglobulins

## Abstract

Colorectal cancer is the source of one of the most common cancer-related deaths worldwide, where the main cause of patient mortality remains metastasis. The aim of this study was to determine the role of CCL7 (chemokine (C-C motif) ligand 7) in tumor progression and finding whether it could predict survival of colorectal cancer patients. Initially, our study focused on the crosstalk between mesenchymal stem cells (MSCs) and CT26 colon carcinoma cells and resulted in identifying CCL7 as a chemokine upregulated in CT26 colon cancer cells cocultured with MSCs, compared with CT26 in monoculture in vitro. Moreover, we showed that MSCs enhance CT26 tumor cell proliferation and migration. We analyzed the effect of CCL7 overexpression on tumor progression in a murine CT26 model, where cells overexpressing CCL7 accelerated the early phase of tumor growth and caused higher lung metastasis rates compared with control mice. Microarray analysis revealed that tumors overexpressing CCL7 had lower expression of immunoglobulins produced by B lymphocytes. Additionally, using Jh mutant mice, we confirmed that in the CT26 model, CCL7 has an immunoglobulin-, and thereby, B-cell-dependent effect on metastasis formation. Finally, higher expression of CCL7 receptor CCR2 (C-C chemokine receptor type 2) was associated with shorter overall survival of colorectal cancer patients. Altogether, we showed that CCL7 is essentially involved in the progression of colorectal cancer in a CT26 mouse model and that the expression of its receptor CCR2 could be related to a different outcome pattern of patients with colorectal carcinoma.

## 1. Introduction

Colorectal cancer is the third most common type of cancer, causing 694,000 deaths in 2012 and accounting for about 10% of all cancer cases [1]. Despite the fact that the survival rates of patients continue to increase substantially, mainly due to improved diagnostics and treatments, 5-year survival still remains under 60% in Europe [2]. The main cause of mortality in colorectal cancer patients is liver metastasis, either present already at the cancer diagnosis stage, or developing after resecting the primary tumor. Therefore, a substantial effort needs to be made in order to identify proteins controlling the metastatic cascade as they have crucial clinical significance in future patient treatments.

Mesenchymal stem cells and the chemokines released by them are a crucial part of the tumor microenvironment and play a role in determining the growth and metastasis of colorectal cancer [3,4]. Our study focused on the chemokine (C-C motif) ligand 7 (CCL7), known also as monocyte-specific chemokine 3 (MCP-3), which attracts monocytes and regulates macrophage function. First, identified from osteosarcoma supernatant [5], it was found to be expressed by different cell types: monocytes, fibroblasts, platelets, colon epithelial cells, and some tumor cell lines [6,7,8]. Due to its capacity to bind to multiple leukocyte receptors, CCL7 affects immune cells and activates monocytes and granulocytes [9].

So far, only a few studies have focused on deciphering the role of CCL7 in cancer development. One of them showed that CCL7 gene transfer to mastocytoma cells caused reduced tumorigenicity, enhanced neutrophil recruitment to the tumor, and dendritic cell infiltration in peritumoral tissue [10]. In another study, CCL7 gene transfection to colorectal cancer cell line resulted in tumor growth retardation and metastasis inhibition [11]. Apart from that, CCL7 and its receptors—CCR1 (C-C chemokine receptor type 1), CCR2, and CCR3 (C-C chemokine receptor type 3)—were found to be overexpressed in liver metastatic tumor tissues of the patients and had much higher expression levels in hepatic recurrences when compared with primary tumors [12]. This was the first clinical report presenting CCL7 as a novel target in liver metastasis of colorectal cancer. However, the mechanisms underlying CCL7 importance in colorectal cancer metastasis have not yet been explained.

In this study, we used a variety of in vitro and in vivo studies in order to discover the role of CCL7 in colorectal cancer progression. In addition, we used a human tissue microarray (TMA) to determine correlations between CCL7 receptors and clinical data from patients.

## 2. Results

### 2.1. Mesenchymal Stem Cells Affect CT26 Tumor Cell Proliferation, Migration, and Expression of CCL7 In Vitro

An in vitro study showed that CT26 colon cancer cells exhibit higher proliferation rates when cultured in direct contact with MSCs (1:1) after 24 h (258 ± 124 pixels per field for CT26 vs. 594 ± 116 for CT26 + MSC, *p* = 0.02), 48 h (467 ± 159 px for CT26 vs. 1343 ± 258 for CT26+MSC, *p* = 0.01), and 72 h (552 ± 112 px for CT26 vs. 3128 ± 1122, *p* = 0.05) (Figure 1A, left upper panel). However, MSCs had no effect on CT26 cells proliferation when cocultured in a transwell setting (Figure 1A, right upper panel), or in the case of using an MSC-conditioned media (*p* > 0.05) (Figure 1A, left lower panel). CT26 cells were also shown to gain increased migration ability once MSCs were used as a chemoattractant (10 ± 5 for CT26 in serum-free media vs. 35 ± 9 cells per field for CT26+MSC, *p* = 0.003) (Figure 1B). In order to find differences in chemokine and chemokine receptors expression between CT26 and MSCs, we employed a DNA microarray. Results showed that MSCs and CT26 cells had different expressions of a variety of chemokines, one of which was CCL7 (+13.63-fold change, upregulation in MSCs vs. CT26) and one of its receptors CCR1 (−5.85-fold change, downregulation in MSCs vs. CT26) (Table 1). Additionally, PCR array analysis confirmed that CCL7 expression was upregulated (+1.46 fold change) and CCR1 downregulated (−1.91-fold change) in cocultured CT26+MSC vs. in CT26 (Table 2). Furthermore, ELISA was performed in order to validate these findings on a protein level. In the case of MSCs cocultured with CT26 in a transwell system, CCL7 concentration was significantly higher when compared to MSCs in monoculture (255 ± 7 vs. 155 ± 15, *p* = 0.002). Similarly, CT26 cocultured with MSCs secreted more CCL7 compared to CT26 in monoculture (77 ± 3 vs. 0, *p* < 0.001) (Figure 1C).

### 2.2. CCL7 Overexpression Accelerates Early Phase of CT26 Tumor Growth In Vivo

A proliferation assay was performed in order to determine whether addition of recombinant CCL7 affected the proliferation rate of CT26 cells in vitro. None of the examined CCL7 concentrations (50, 100, and 200 ng/mL) caused a change in the proliferation of CT26 cells (*p* > 0.05) (Figure 2A). Next, CCL7 overexpression in the mCCL7+ cell line was confirmed using ELISA (0.08 ± 0.05 in blank control vs. 2.39 ± 0.21 in mCCL7+, *p* = 0.03) (Figure 2B). Additionally, no difference in the proliferation rate was observed between mCCL7+ and blank control cells (Figure 2C). Interestingly, a migration assay did not show a statistically significant difference in migration between two analyzed cell lines assessed via scratch assay (*p* > 0.05) (Figure 2D).

### 2.3. Overexpressed CCL7 Enhances Lung Metastasis Formation In Murine CT26 Tumor Model

Finally, an in vivo study revealed that subcutaneous injection of CT26 cells overexpressing CCL7 led to acceleration in the early phase of tumor growth (between day 5 to day 11) when mCCL7+ tumor-bearing mice had bigger tumor size when compared with blank control-injected mice (Figure 2E). However, after day 11, tumors in both groups of mice did not differ statistically significantly in size (*p* > 0.05).

As no striking morphological differences were observed between the tumors (Figure 3A,B), murine organs including lung, liver, and brain were collected and analyzed in order to study the effect of CCL7 overexpression on metastasis formation. Morphological analysis of the lungs (Figure 3C,D) revealed that mCCL7+ cell implantation led to metastases in 73% of the mice, compared with 40% of control mice developing metastatic foci, which indicated that overexpression of CCL7 promoted metastatic spread of CT26 tumors to the lungs (Figure 3E). Moreover, lungs from CCL7-overexpressing tumor-bearing mice had severe intra-alveolar bleedings, which were absent in the case of the control mice.

### 2.4. CCL7 Overexpression in CT26 Tumors Suppresses Expression of Immunoglobulins

In order to unravel the mechanism of how CCL7 affects metastasis formation, a microarray analysis was performed using RNA extracts from CT26 cells overexpressing CCL7 or expressing the control vector. Gene profiling revealed a set of multiple genes having statistically different (*p* < 0.05) expression level in blank control vs. mCCL7+ cells (Table 3). The analysis carried out by Panther DB service (Los Angeles, CA, USA) indicated that, among the genes mentioned, genes encoding proteins involved in inflammation were the most significant group compared with gene groups related to other pathways and included genes encoding MHCIIβ, complement C4a precursor, and immunoglobulin heavy chains (Ighv7183, Ighv8-5). To find out whether a similar pattern of gene expression could be observed in vivo, tumors dissected from mice were also profiled via microarray analysis, using RNA extracts from CT26 tumors overexpressing CCL7 (*n* = 3), or control vector (*n* = 3) (Figure 4A). In this setup, 1632 genes had statistically different (*p* < 0.05) expression levels in blank control vs. mCCL7+ tumors. Results were further analyzed with Panther DB service, and similar to previous findings on cells, the inflammation-related group of genes was the most representative. Interestingly, genes related to B cell maturation were the ones most affected. Namely, CCL7 overexpressing tumors had a significantly lower expression of multiple genes encoding for immunoglobulin heavy and light chains. This result suggested that CCL7 overexpression suppresses immunoglobulins expression, hence affecting B cell development and maturation. To verify whether B cell maturation is crucial in the CCL7-dependent mechanism of metastasis formation, we used Jh constitutive mutant mice (Taconic Biosciences, Rensselaer, NY, USA), which carry a deletion of the endogenous murine J segments of the heavy Ig heavy chain locus and have no mature (immunoglobulin-bearing) B cells. Jh mice were injected with a blank control or mCCL7+ CT26 cells. No differences in the tumor growth were observed between the groups (Figure 4B). Interestingly, 80% of mCCL7+ CT26-injected tumor-bearing Jh mice developed lung metastasis, compared to 100% of mice in the blank control group (Figure 4C). However, the total number of lung metastases per mouse (Figure 4D) and average number of lung metastases per section (Figure 4E) were similar in both groups. This data confirmed that CCL7 had an immunoglobulin-dependent effect on metastasis formation in the B-cell-competent CT26 model, as opposed to B-cell-deprived Jh mice.

### 2.5. CCR2 Expression Predicts Overall Survival of Colorectal Cancer Patients

In order to check expression level of CCL7 receptors in CT26-derived cell lines (blank control and mCCL7+), real time PCR was performed. The analysis showed that the mRNA level of CCR1 was lower in the mCCL7+ cell line when compared with blank control cell line (1 in blank control vs. 0.72 ± 0.09 in mCCL7+, *p* = 0.04), whereas mRNA level of CCR2 was higher in mCCL7+ vs. blank control (1 in blank control vs. 1.9 ± 0.17 in mCCL7+, *p* = 0.02) (Figure 5A). However, no CCR3 mRNA was detected in the studied cells. Immunohistochemical stainings for CCL7 receptors were performed on tissue microarrays (TMA) containing colorectal cancer patient samples in order to determine the relationship between CCL7 receptor expression and clinical patient characteristics, such as patient survival. Our analysis showed that colorectal cancer patients exhibited different grades of CCL7 receptors expression, assessed as low (Figure 5B) or high (Figure 5C) by a pathologist. Descriptive data analysis was performed in order to determine whether any of the analyzed CCL7 receptors were associated with survival of the patients (Table 4). Interestingly, 73% of patients who developed lymph node metastases had a high expression of CCR2, compared with only 27% of metastasis-positive patients having lower CCR2 expression. Second, high CCR2 expression in patient samples was associated with shorter relapse-free survival (RFS) of colorectal cancer patients (981 ± 865 days) compared with low expression (1403 ± 1279 days) (Table 4). Moreover, high CCR2 expression was also associated with shorter overall survival (OS) of colorectal cancer patients (1219 ± 988 days) compared with low expression (1622 ± 1197 days) (Table 4). This result was confirmed with Kaplan–Meier analysis (Figure 5D) (*p* < 0.05). Altogether, these data indicate that the high expression of CCR2 was significantly associated with OS. Conversely, no associations of CCR1 or CCR3 with patient survival were discovered.

## 3. Discussion 

In this study, we report that MSCs enhanced CT26 tumor cell proliferation and migration. However, it is not the first report on the influence of MSCs on tumor progression, especially in colon cancer. Previously, MSCs were shown to modulate tumorigenicity of colon cancer through interleukin-6 (IL-6) [13] and promotes the formation of colorectal tumors in mice [14]. Moreover, MSCs enhance the growth and metastasis of colon cancer by promoting angiogenesis, migration, and invasion, and by inhibiting tumor cell apoptosis [15].

Our results identified CCL7 as a chemokine upregulated in CT26 colon cancer cells cocultured with MSCs compared with CT26 in monoculture in vitro. Moreover, our animal experiments indicated that overexpression of CCL7 promoted the metastatic spread of colon CT26 tumors to the lungs. Previously, there has not been many studies published analyzing the effect of CCL7 expression on metastasis formation in the colorectal cancer model. In one of them, Hu et al. showed that CCL7 gene transfection to a colorectal cancer cell line leads to tumor growth retardation and metastasis inhibition in subcutaneously inoculated mice [11]. However, the discrepancy between his group’s results and ours can be explained in many ways. For example, in our study, we used overexpression of CCL7 in CT26 tumor model and looked at metastasis in the lungs, while Hu et al. employed a model of CCL7 gene transfection to mouse rectal cancer CMT93 cell line and observed metastasis in the lymph nodes. On the other hand, in another study on colon cancer patients, CCL7 overexpression and its receptor expression was correlated with metastasis to the liver [12]. Similarly, a study on BALB/c nu/nu mice injected directly into cecum wall with hCCL7 overexpressing human colorectal HCT116 cells revealed metastasis formation both in the lungs (4/5) and in the liver (3/5); therefore, the type of cells, origin, and site of cancer cells inoculation (ectopic/subcutaneous or orthotropic/directly in colon wall) might be significant for the tropism of colon cancer cells to different organs [16].

Interestingly, several other research groups showed that CCL7 expression can be associated with enhanced invasiveness and tumor metastasis in other types of cancer. For example, CCL7 was reported to be upregulated in cancer-associated fibroblasts (CAFs) cultured with oral squamous cell carcinoma (OSCC) cells and potentiated tumor migration and invasion in vitro [17]. Furthermore, CCL7 overexpression was associated with lymph node metastasis and poor prognosis in gastric cancer [18]. Even brain cancer metastases were shown to carry a higher expression of CCL7 compared with primary tumors of renal cell cancer (RCC) [19]. Finally, recent studies confirmed the role of the COX2-MMP1/CCL7 axis in brain cancer metastasis in breast cancer [20].

Even though there is no previous evidence in the literature for CCL7 abolishing immunoglobulin expression specifically during cancer progression, it was reported that MSC–derived CCL2 suppresses plasma cell immunoglobulin production via STAT3 inactivation and PAX5 induction [21]. Although our study does not provide detailed information about how CCL7 suppresses the expression of immunoglobulins, it is an exciting finding that requires further examination.

Our tissue microarray (TMA) analysis revealed that CCR2 is a predictive factor for the overall survival (OS) of colorectal cancer patients. It is the first report of such a role for this receptor. Previously, only CCL2–CCR2 signaling has been shown to inhibit breast cancer metastasis in vivo [22]. Interestingly, other CCL7 receptors have been demonstrated to affect liver metastasis. Namely, CCR1 was shown to mediate the accumulation of myeloid cells in the liver microenvironment, enhancing mouse colon cancer metastasis [23,24], and additionally, to affect liver metastasis and angiogenesis in a thymoma model [25]. Altogether, our study identified CCL7 as a factor affecting lung metastasis in CT26 colon cancer model and its receptor, CCR2, as a predictor of the overall survival of colorectal cancer patients. Even though our study does not provide a strong link between the in situ expression pattern of CCL7 and CCR2, it sets the stage for future studies analyzing CCL7/CCR2 axis in colorectal cancer progression. Moreover, further research needs to be done in order to decipher the immunoglobulin-dependent mechanism, through which CCL7 controls metastasis formation and to confirm the role of the CCR2–CCL7 axis in this process.

## 4. Materials and Methods

### 4.1. Cell Lines

The mouse colon carcinoma CT26.CL25 cell line was obtained from American Type Culture Collection (ATCC, Manassas, VA, USA) and used for in vitro studies and introducing lentivirus-mediated CCL7 overexpression. The cell line marked as blank control contained an empty vector, while the one marked as CCL7+ carried vector overexpressing CCL7. All cell lines were cultured in RPMI 1640 medium (Gibco/ThermoFisher, Waltham, MA, USA ) with 2 mM L-glutamine adjusted to contain 1.5 g/L sodium bicarbonate, 4.5 g/L glucose, 10 mM HEPES, and 1.0 mM sodium pyruvate, and supplemented with 0.1 mM non-essential amino-acids and 0.4 mg/mL G418, 10% fetal bovine serum, and 1% penicillin and streptomycin in a humidified incubator at 37 °C in an atmosphere containing 5% CO_2_ and 95% O_2_.

### 4.2. MTT Proliferation Assay

CT26.CL25, blank control, or CCL7+ cells were seeded in 96-well plates with a density of 2500 cells/well. In the case of CT26.CL25, different concentrations of CCL7 (50, 100, and 200 ng/mL) were added to the medium. Additionally, wells with the culture medium without the cells were used as a negative control for the experiment. After 1, 3, 5, and 7 days, plates were analyzed for proliferation with an MTT assay. MTT solution (5 mg/mL MTT in PBS) was added to each well and incubated for 4 h at 37 °C. Then the media were removed and the MTT solvent (4 mM HCl, 0.1% Nonidet P-40 (NP40) (Sigma Aldrich, Saint Louis, MO, USA) in isopropanol) was added. Plate was covered with aluminium foil, agitated on a shaker for 15 min, and then the absorbance was read using a spectrophotometer (Fisher Scientific, Hampton, NH, USA) at 590 nm with a reference filter of 620 nm.

### 4.3. Migration Assays

Migration of tumor cells was evaluated using a QCM Chemotaxis Cell Migration Kit (Millipore, Burlington, MA, USA) according to the manufacturer’s protocol. In this assay, 24-well transwell chambers and inserts with 8 µm pores were used.

Alternatively, migration was assessed with a scratch assay. Briefly, confluent cells monolayers were scratched with a pipette tip. Media were replaced and wounds were photographed after 24 h. Image J (National Institutes of Health, Bethesda, Maryland) was used to measure closure of the wound. Six individual measurements of wound size for each wound were averaged and experiments involved three replicates per experiment.

### 4.4. Microarray Analysis

Cell pellets were trypsinized, and tumor tissue was frozen immediately after excision from the animals. Material was subsequently used for RNA extraction with RNeasy Mini Kit (Qiagen, Hilden, Germany) according to the manufacturer’s instructions and subsequent microarray analysis. Cell pellets and well as three tumors tissue samples were used for analyzing each group. Gene expression profiles were analyzed using an Affymetrix (Santa Clara, CA, USA) (MSCs vs. CT26 cells) or MoGene (Saint Louis, Missouri, USA) array (blank control vs. mCCL7 + cells and tumors). A heatmap was created using online software Heatmapper (Basel, Switzerland) [26]. Gene expression profiles were additionally analyzed with the Panther DB service (University of Southern California, Los Angeles, CA, USA).

### 4.5. ELISA

Cells were seeded on a 24-well plate with 70% confluency. Their medium was changed soon after they adhered and cells were then cultured for 2 days in previously described conditions. A total of 500 μL of cell supernatants were collected and centrifuged at 1500 rpm for 5 min. Subsequently, pellets were discarded and supernatants used for Mouse CCL7 ELISA (Biorbyt, Cambridge, UK), according to the manufacturer’s protocol.

### 4.6. Mouse Tumor Models

Approximately 3 × 10^6^ tumor cells (blank control or CCL7+) were subcutaneously implanted in the dorsal back of 6- to 8-week-old female wild type BALB/c (*n* = 8 per group) and tumor sizes were measured with a caliper every 2 days, starting from day 5 after implantation. To study metastasis, primary tumors were surgically removed under analgesia and anesthesia when they reached around 1000–1500 mm^3^. The experiment was continued for two months after tumor excision. Subsequently, mice were sacrificed via cervical dislocation and their organs were collected for further analysis. The same procedures were employed for Jh constitutive knockout mice.

Wild type BALB/c mice were obtained from Harlan Laboratories (Indianapolis, IN, USA) and Jh mutant mice were obtained from Taconic Biosciences. All animal studies were performed in Institute of Genetics and Animal Breeding (Jastrzebiec, Poland) in accordance with international guidelines and Polish law, and approved by the II Local Ethics Committee (permit number WAW/184/2018).

### 4.7. Immunohistochemical Stainings

Tumors, as well as other murine tissues (brain, liver, lungs), were fixed in 4% paraformaldehyde for 24 h and embedded in paraffin. Sections (5 μm) were cut, deparaffinized in Tissue Clear (Sakura Finetek Europe, Alphen aan den Rijn, The Netherlands) and hydrated in a descending alcohol series. Subsequently, sections were stained with hematoxylin and eosin in order to perform morphological analysis of the tissue.

TMAs were also stained with CCR1, CCR2, and CCR3 antibodies (AbCam, Cambridge, UK). Antigen retrieval was obtained by heating the sections in Citra Plus Solution (Biogenex, Fremont, CA, USA) for 20 min in the microwave. After cooling down, sections were washed with PBS and sequentially incubated with 3% hydrogen peroxidase (Merck, Darmstadt, Germany) for 30 min at room temperature to block nonspecific binding and with 5% serum for 1 h. Then, the slides were incubated with primary antibodies overnight at 4 °C. Positive cells were detected with VectaStain Solution (Vector Laboratories, Burlingame, CA, USA). Briefly, slides were incubated with a biotinylated anti-rabbit antibody for 1 h and then with a Vectastain detection system horseradish peroxidase-labeled streptavidin. Antigens were visualized with diaminobenzidine (Innovex Biosciences, Richmond, CA, USA). Then, the slides were counterstained with hematoxylin (Vector Laboratories, Burlingame, CA, USA) and mounted with Permount (Dako/Agilent Technologies, Santa Clara, CA, USA) mounting medium.

### 4.8. Quantitative Real-Time RT-PCR

Total RNA was extracted from the cells using an RNeasy Mini Kit (Qiagen) according to the manufacturer’s instructions. Amplification of mRNA was performed and then it was transcribed from double-stranded cDNA using SuperScript™ III Reverse transciptase (Invitrogen, Carlsbad, CA, USA).

Quantitative real-time RT-PCR was performed in triplicate in 96-well plates; each 10 μL reaction consisted of 5 μL of FAST Universal Master Mix (Applied Biosystems, Foster City, CA, USA), 0.5 μL of probe primer sets of CCR1 (Life Technologies, Carlsbad, CA USA), CCR2 (Life Technologies), CCR3 (Life Technologies), or ACTB (Applied Biosystems), 1.5 μL of ddH20, and 3 μL of template.

The real-time PCR analysis was performed on an Applied Biosystems Prism 7900 Sequence Detection System (Applied Biosystems).

Ct values were normalized for the deviations against *ACTB* serving as a housekeeping gene. The differential gene expression was estimated as: ΔCt = Ct_(blank control)_ − Ct_(mCCL7+)_, and fold-change = 2^(−ΔCt)^.

### 4.9. Clinical Samples of Human Colon Cancer

Tissue microarrays (TMA) containing 276 samples of human colorectal cancer were commercially available and obtained from Yale University School of Medicine (New Haven, CT, USA). The study was done according to the principles embodied in the Declaration of Helsinki.

### 4.10. Statistical Analysis

Human studies: Survival curves were calculated for the various groups using the Kaplan–Meier method and compared by the log-rank test. SPSS Statistic version 22 software (IBM, Armonk, NY, USA) was used for the analysis.

Animal studies: Differences in obtained values between the groups (blank control and mCCL7+) were assessed with *t*-test, where *p* < 0.05 was considered statistically significant.

## Figures and Tables

**Figure 1 ijms-20-00686-f001:**
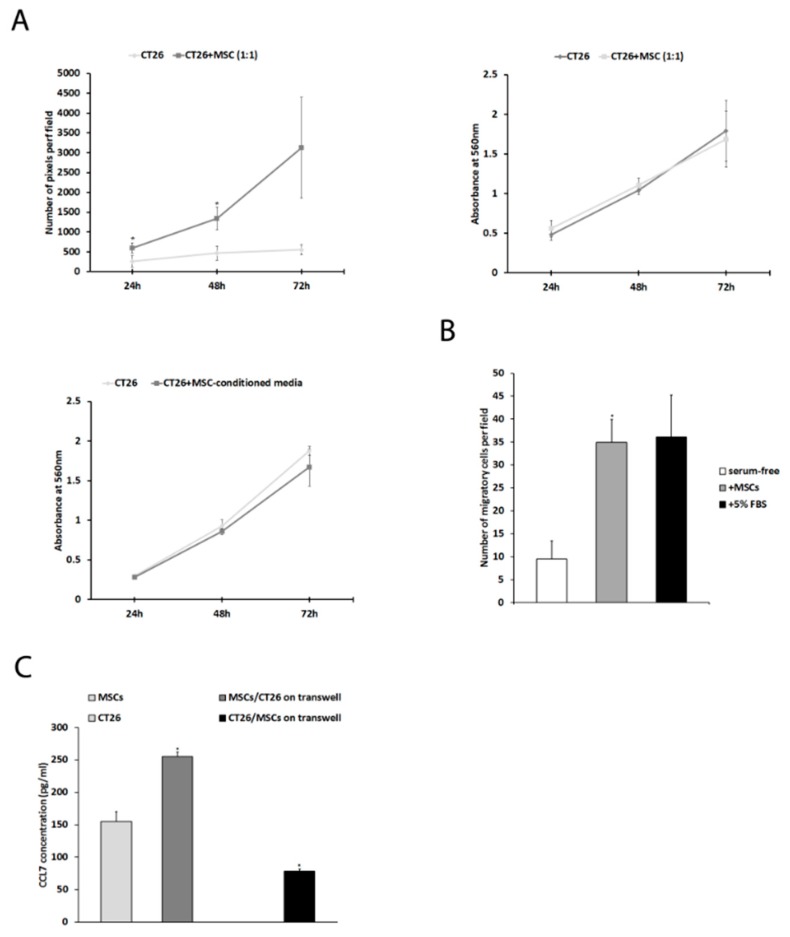
Mesenchymal stem cells affect CT26 tumor cells proliferation, migration, and expression of CCL7: (**A**) proliferation rate of CT26 cells cocultured with MSC on the same dish (1:1, upper left panel), with MSC in a transwell setup (1:1, upper right panel), and with MSC-conditioned media (lower left panel); (**B**) effect of MSCs on migration of CT26 cells; (**C**) CCL7 concentration measured by ELISA in different setups: only CT26, only MSCs, MSC/CT26 transwell, and CT26/MSC transwell.

**Figure 2 ijms-20-00686-f002:**
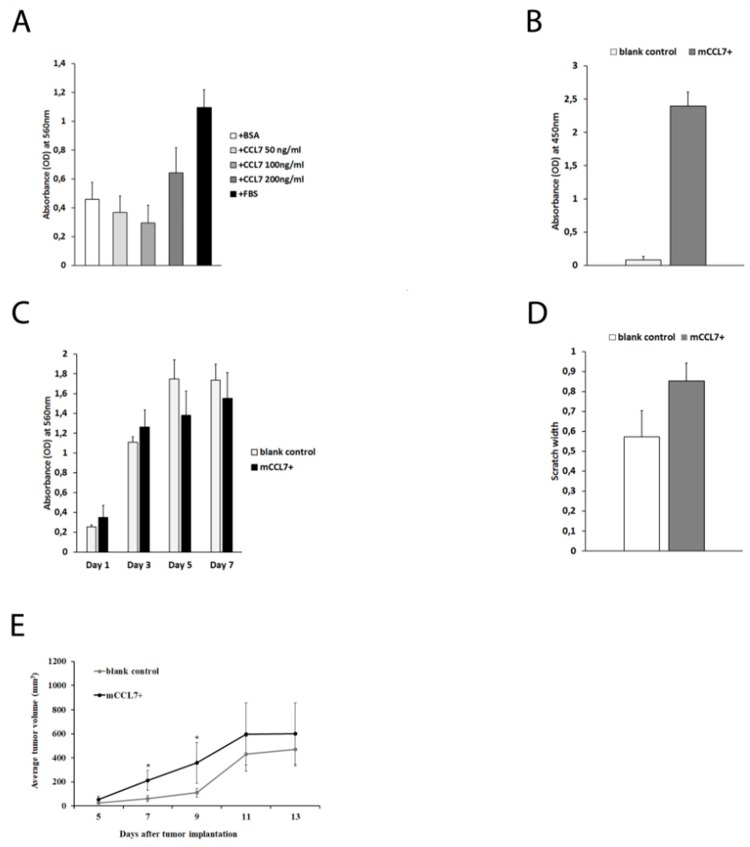
Effect of CCL7 overexpression on tumor growth: (**A**) proliferation of CT26 cells subject to recombinant CCL7; (**B**) overexpression of CCL7 in mCCL7+ cell line (ELISA for CCL7); (**C**). CT26 tumor cell proliferation (blank control vs. mCCL7+) (MTT assay); (**D**) scratch assay on same cell lines; and (**E**) CT26 tumor growth (blank control vs. mCCL7+) in BALB/c mice (*n* = 15 mice per group).

**Figure 3 ijms-20-00686-f003:**
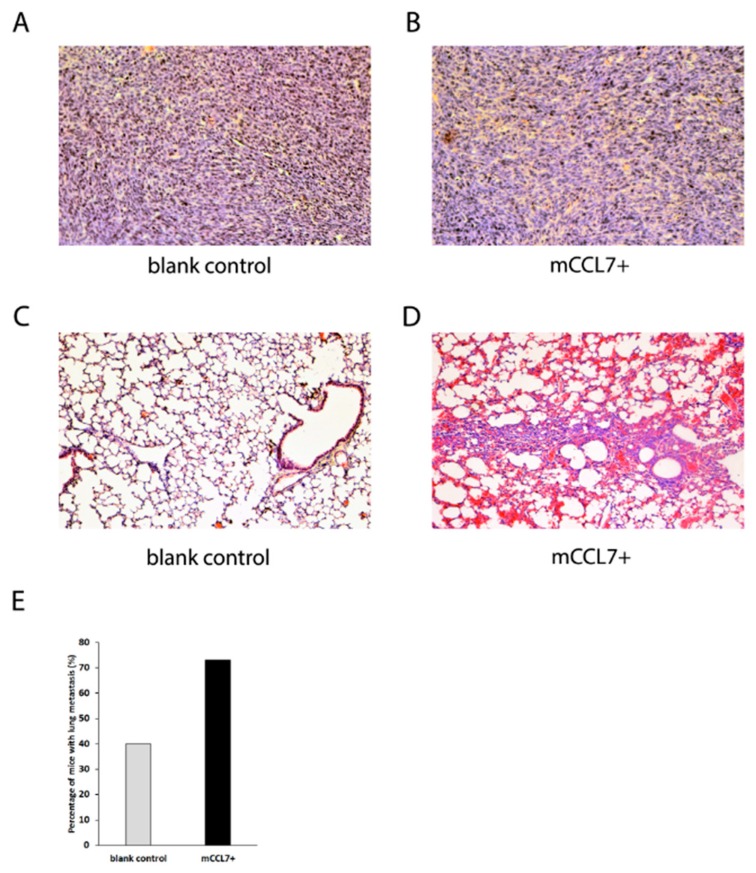
CCL7 affects lung morphology and metastasis in vivo: (**A**) tumors stained for hematoxylin and eosin (blank control); (**B**) tumors stained for hematoxylin and eosin (mCCL7+); (**C**) lungs stained for hematoxylin and eosin (blank control); (**D**) lungs stained for hematoxylin and eosin (mCCL7+); and (**E**) percentage of tumor-bearing mice developing lung metastasis (*n* = 6–8 mice per group, 8 sections per mouse).

**Figure 4 ijms-20-00686-f004:**
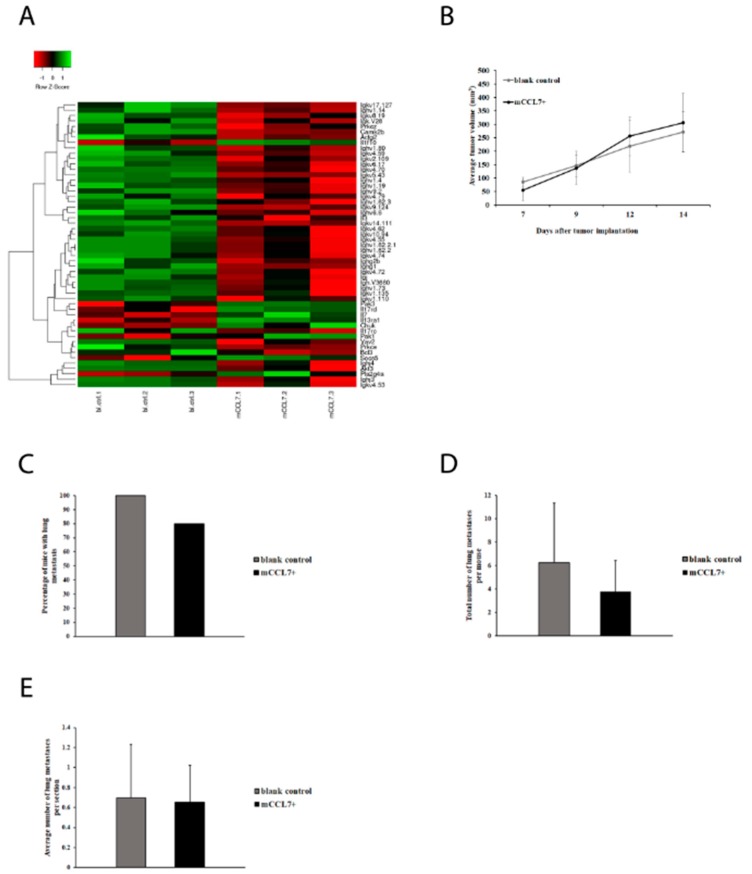
CCL7 affects lung metastasis via immunoglobulin-dependent mechanism in B-cell competent mice, but not B-cell-deprived Jh mice: (**A**) microarray analysis of CT26 tumors, (**B**) CT26 tumor growth in Jh mutant mice (*n* = 10 mice per group), (**C**) percentage of tumor-bearing Jh mice with lung metastasis, (**D**) total number of lung metastases per mouse (*n* = 8 mice per group), and (**E**) average number of lung metastases per lung section (*n* = 8 sections per mouse).

**Figure 5 ijms-20-00686-f005:**
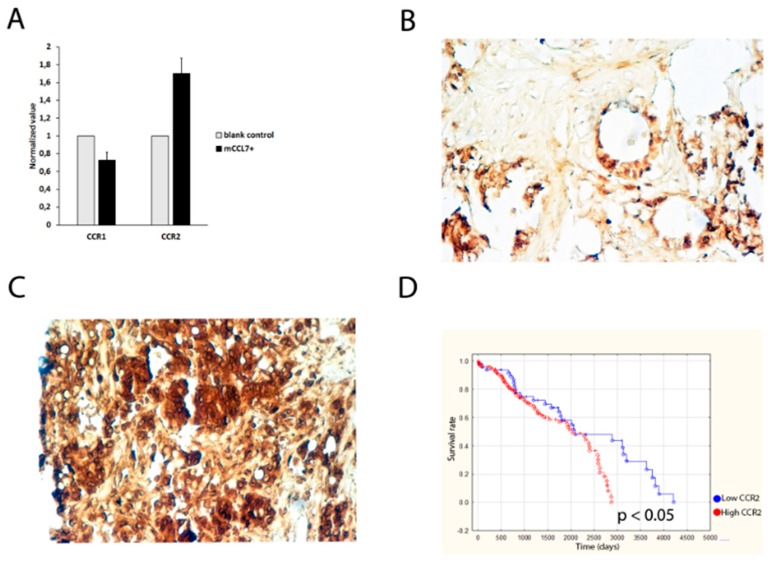
CCR2 expression in cells and colon cancer patients: (**A**) CCL7 receptor expression assessed using real time PCR, (**B**) low CCR2 expression in colon cancer patients, (**C**) high CCR2 expression on TMA, and (**D**) Kaplan–Meier plot of overall survival (OS) for patients stratified according to CCR2 expression assessed as low or high.

**Table 1 ijms-20-00686-t001:** Comparison of gene expression patterns of chemokine and chemokine receptors obtained by DNA microarray on CT26 and MSCs.

Gene Symbol	Fold Change (CT26 vs. MSC)	Regulation (CT26 vs. MSC)
Ccl2	−18,087187	down
Ccl5	−20,740484	down
Ccl7	−13,637749	down
Ccl8	−6,8785295	down
Ccl27a	−19,154331	down
Ccr1	5,850982	up
Ccrl2	−6,921777	down
Cxcl1	−13,09386	down
Cxcl12	−877,7667	down
Cxcl13	13,571245	up
Cxcl16	−47,79881	down
Cxcl16	−24,529097	down
Cxcl2	−35,86758	down
Cxcl3	−6,8512273	down
Cxcl5	−83,25011	down
Cxcr7	−6,042842	down
Ccl2	−18,087187	down

**Table 2 ijms-20-00686-t002:** Regulation of chemokines and chemokine receptors using a PCR array on cocultured CT26 and MSCs.

Gene Symbol	Fold Change (CT26 + MSC vs. CT26)
C5ar1	1,1
Ccbp2	1,1
Ccl1	1,1
Ccl11	−1,03
Ccl12	1,1
Ccl17	1,58
Ccl19	1,1
Ccl2	2,07
Ccl20	2,97
Ccl22	1,1
Ccl24	1,1
Ccl25	1,21
Ccl26	1,1
Ccl28	1,1
Ccl3	1,1
Ccl4	1,93
Ccl5	1,82
Ccl6	3,12
Ccl7	1,46
Ccl8	1,25
Ccl9	2,04
Ccr1	−1,91
Ccr10	−1,09
Ccr1l1	1,1
Ccr2	1,1
Ccr3	1,1
Ccr4	1,1
Ccr5	1,1
Ccr6	1,1
Ccr7	1,1
Ccr8	1,1
Ccr9	1,1
Ccrl1	1,1
Ccrl2	1,1
Cmklr1	1,1
Cmtm2a	1,1
Cmtm3	−1,07
Cmtm4	1,05
Cmtm5	1,1
Cmtm6	1,25
Cx3cl1	1,1
Cx3cr1	1,32
Cxcl1	−1,43
Cxcl10	1,87
Cxcl11	1,1
Cxcl12	1,1
Cxcl13	−1,48
Cxcl14	1,1
Cxcl15	1,15
Cxcl16	1,11
Cxcl2	1,1
Cxcl3	1,1
Cxcl5	−2,21
Cxcl9	1,1
Cxcr1	1,1
Cxcr2	1,1
Cxcr3	1,1
Cxcr4	−1,03
Cxcr5	1,1
Cxcr6	1,19
Cxcr7	1,32
Darc	1,74
Fpr1	1,1
Gpr17	1,1
Hif1a	1,19
Ifng	1,1
Il16	1,1
Il1b	1,1
Il4	1,1
Il6	1,51
Itgam	1,1
Itgb2	1,04
Mapk1	1,17
Mapk14	1,08
Pf4	1,21
Ppbp	−1,78
Slit2	−1,13
Tgfb1	1,07
Tlr2	1,1
Tlr4	−1,31
Tnf	1,1
Tymp	1,03
Xcl1	1,1
Xcr1	1,1

**Table 3 ijms-20-00686-t003:** Comparison of gene expression patterns of CCL7-overexpressing CT26 cells (mCCL7+) vs. blank control cells. Table depicts top upregulated genes.

Gene Symbol	Fold Change (mCCL7+ vs. Blank Control)	Regulation (mCCL7+ vs. Blank Control)
Obox2	1.890295359	up
H2-DMb2	1.765182186	up
C4a	1.674796748	up
Olfr102	1.603092784	up
Olfr907	1.547263682	up
Mcpt4	1.546511628	up
Rhox2h	1.529680365	up
Igh-V7183	1.520408163	up
Akr1c19	1.495283019	up
Ccl7	1.495238095	up
Scgb2b6	1.478723404	up
Olfr347	1.464539007	up
Vmn2r28	1.458715596	up
Ighv8-5	1.458563536	up
Fpr2	1.457413249	up

**Table 4 ijms-20-00686-t004:** Characteristics of colorectal cancer patients stratified according to low and high grading of CCR2 expression.

CCR2 Expression	Low	High
number of patients	56	149
female	29	80
male	27	69
age (years) ± SD	68 ± 15	69 ± 12
LN metastasis + (% of positive patients)	25 (27%)	69 (73%)
RFS (days) ± SD	1403 ± 1279	981 ± 865
OS (days) ± SD	1622 ± 1197	1219 ± 988

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
