# Peer review of "Effect of Chemokine (C-C Motif) Ligand 7 (CCL7) and Its Receptor (CCR2) Expression on Colorectal Cancer Behaviors"

_ijms, 2019, doi:10.3390/ijms20030686_

Round 1

Reviewer 1 Report

Specific comments to the authors:

The authors Ewa Kurzejamska et al. study the role of Chemokine (C-C motif) ligand 7 (CCL7) and its receptor CCR2 in colorectal cancer in vivo, in vitro and in situ. Based on their applied molecular techniques the authors could demonstrate that (i) CCL7 is essentially involved in the progression of colorectal cancer in an in-vivo-modell and that (ii) the expression of its receptor CCR2 could be related to different outcome pattern of patients with colorectal carcinoma.

Overall, the manuscript is well done, easy to follow and to understand. The methods are overall well described. Although the results are clear presented, some minor concerns limit the impact of the manuscript (see specific comments). Finally, the discussion should discuss more, how the findings could be possibly used for therapeutic options. The three limitations of the study mentioned in the specific comments in detail must be more emphasized by the authors.

In conclusion, the presented data are interesting. After incorporating the mentioned specific comments (see below) the manuscript has the potency to be accepted.

Specific comments

Abstract: Please add a definitive conclusion of the findings at the end of the abstract.

Introduction: Please add some literature dealing the integrative interaction of colorectal cancer, mesenchymal stem cells and chemokines.

Results: Please add the confidence intervals to all figures with quantitative data. Please add the significant differences between CCR2 expression group low and high in the table 4. Please add the p-values of the log-rank test in figure 5D.

Discussion: Follwing two limitations must be discussed by the authors more in detail: i) The definitive link between the in-situ-expression pattern of CCL7 and CCR2 is not shown. (ii) The definitive pathomechanism of CCR2 in-situ is not analyzed by the authors resulting in more questions than answers. (iii) How could the findings of this animal-model and in-situ-analysis transferred to clinical issues in future?

Presentation: Overall, the presentation is good.

Author Response

Response to Reviewer 1 Comments

Dear Reviewer,

Thank you so much for your constructive comments and suggestions. I tried to address them as well as I could and I will be waiting for your feedback. I hope that the corrections which I introduced to the manuscript will increase its scientific value.

Best regards,

Ewa Kurzejamska

Abstract: Please add a definitive conclusion of the findings at the end of the abstract.

I added the final conclusion to the abstract, mentioning that CCL7 is essentially involved in the progression of colorectal cancer in CT26 mouse model and that the expression of its receptor CCR2 could be related to different outcome pattern of patients with colorectal carcinoma.

Introduction: Please add some literature dealing the integrative interaction of colorectal cancer, mesenchymal stem cells and chemokines.

I added a brief sentence, where I underlined the impact of MSCs and chemokines in colorectal cancer progression, giving appropriate examples from the literature.

Results: Please add the confidence intervals to all figures with quantitative data. Please add the significant differences between CCR2 expression group low and high in the table 4. Please add the p-values of the log-rank test in figure 5D.

I added confidence intervals to all the figures with quantitative data, like graphs including tumor growth or proliferation studies. I also added requested p value in Fig.5D.

Discussion: Follwing two limitations must be discussed by the authors more in detail: i) The definitive link between the in-situ-expression pattern of CCL7 and CCR2 is not shown. (ii) The definitive pathomechanism of CCR2 in-situ is not analyzed by the authors resulting in more questions than answers. (iii) How could the findings of this animal-model and in-situ-analysis transferred to clinical issues in future?

I fully agree that we do not show a definitive link between the in-situ-expression pattern of CCL7 and CCR2, which is one of the main drawbacks of our study. It is also hard to compare the results from in vitro study using mouse CCL7 and its receptors with analysis on human samples. Of course we cannot exclude the possibility, that in human samples the link between CCL7 and CCR2 expression is not as strong/evident as in our in vitro mouse experiment.  We did not have a chance to explore this link on a deeper level in human studies. Nevertheless, it is interesting that high expression of CCR2, as a receptor for CCL7, was correlated with shorter RFS and OS, as well as lymph node metastasis of colorectal cancer patients. This information might set the stage for future studies looking at CCL7/CCX2 in colorectal cancer progression, which I briefly mentioned in the discussion.

Reviewer 2 Report

I acknowledge your efforts to accomplish the study, and hope that my comments would improve the value of your study,

1. One of your conclusions that CCL7 overexpression in tumor cells could promote lung metastasis in mice is based on the difference in rates of tumor cells on microscopic examination (73% versus 40%). Please show the sample size and actual P value in the text

2. For better understandings, I recommend that you replace Table 4 with a Figure, in which crude or adjusted survival curve according to the CCR2 status would be shown.

3. As you know, monocyte chemoattractant proteins could have either growth-promoting or growth-inhibiting influences on cancer cells, and I think that solid conclusions cannot be drawn from your study. Therefore, as the title, “Effect of chemokine (C-C motif) ligand 7 (CCL7) and its receptor (CCR2) expression on colorectal cancer behaviors” may be more feasible.

Author Response

Response to Reviewer 2 Comments

Dear Reviewer,

Thank you so much for your constructive comments and suggestions. I tried to address them as well as I could and I will be waiting for your feedback. I hope that the corrections which I introduced to the manuscript will increase its scientific value.

Best regards,

Ewa Kurzejamska

1. One of your conclusions that CCL7 overexpression in tumor cells could promote lung metastasis in mice is based on the difference in rates of tumor cells on microscopic examination (73% versus 40%). Please show the sample size and actual P value in the text.

This result was based on calculating percentage of mice in each group that had micrometastases, not the amount of tumor cells. I added sample size – on how many mice was that calculated and how many sections were assessed per mouse. However, as it was mostly qualitative and rather descriptive analysis (yes or no for presence of metastasis), I could only calculate percentage of mice that had metastases, but could not get in this case confidence intervals or p value.

2. For better understandings, I recommend that you replace Table 4 with a Figure, in which crude or adjusted survival curve according to the CCR2 status would be shown.

I am not sure if I understood your comment right, because I have already included a figure (Fig. 5D) with survival curves according to the CCR2 status of the colorectal cancer patients. I added now a p value to this figure as well. However, Table 4 describes characteristics of colorectal cancer patients stratified according to low and high grading of CCR2 expression. I think it includes important information regarding those patients, and this kind of data is normally presented in a form of a table rather than a graphic figure.

3. As you know, monocyte chemoattractant proteins could have either growth-promoting or growth-inhibiting influences on cancer cells, and I think that solid conclusions cannot be drawn from your study. Therefore, as the title, “Effect of chemokine (C-C motif) ligand 7 (CCL7) and its receptor (CCR2) expression on colorectal cancer behaviors” may be more feasible.

I agree that the title which I proposed was confusing and thank you for your proposal on how to change it. I have changed the title according to your advice and believe that it is now more feasible and understandable for the manuscript’s readers.